# Studies of rice *Hd1* haplotypes worldwide reveal adaptation of flowering time to different environments

Cheng-Chieh Wu[1,2], Fu-Jin Wei[1¤], Wan-Yi Chiou[1], Yuan-Ching Tsai[3], Hshin-Ping Wu[1], Dhananjay Gotarkar[1], Zhi-Han Wei[3], Ming-Hsin Lai[4], Yue-le Caroline Hsing[1,5]*

**1** Institute of Plant and Microbial Biology, Academia Sinica, Taipei, Taiwan, **2** Institute of Plant Science, National Taiwan University, Taipei, Taiwan, **3** Department of Agronomy, National Chia-yi University, Chiayi, Taiwan, **4** Crop Science Division, Taiwan Agriculture Research Institute, Taichung, Taiwan, **5** Department of Agronomy, National Taiwan University, Taipei, Taiwan

¤ Current address: FourOneFour Studio, Taoyuan, Taiwan
* bohsing@gate.sinica.edu.tw

**Data Availability Statement:** All relevant data are within the manuscript and its Supporting Information files.

## Abstract

Rice domestication/adaptation is a good model for studies of the development and spread of this important crop. Mutations that caused morphological and physiological change, followed by human selection/expansion, finally led to the improvement of phenotypes suitable for different kinds of environments. We used the sequence information for *Heading date 1* (*Hd1*) gene to reveal the association between sequence changes and flowering phenotypes of rice in different regions. Seven loss-of-function *hd1* haplotypes had been reported. By data-mining the genome sequencing information in the public domain, we discovered 3 other types. These loss-of-function allele haplotypes are present in subtropical and tropical regions, which indicates human selection. Some of these haplotypes are present locally. However, types 7 and 13 are present in more than one-third of the world's rice accessions, including landraces and modern varieties. In the present study, phylogenetic, allele network and selection pressure analyses revealed that these two haplotypes might have occurred early in Southeastern Asia and then were introgressed in many local landraces in nearby regions. We also demonstrate that these haplotypes are present in weedy rice populations, which again indicates that these alleles were present in rice cultivation for long time. In comparing the wild rice sequence information, these loss-of-function haplotypes occurred *in agro* but were not from wild rice.

## Introduction

Rice (*Oryza sativa*) is one of the most important crops in the world. It is a short-day (SD) plant and was domesticated originally in a temperate zone, then brought to subtropical and tropical zones with warmer temperature and different photoperiod. One of the main reasons for the spread of rice cultivation to a wide range in Asia as well as the increase in production is the diversification of flowering time [1].

**Funding:** This project was supported by funding from Ministry of Science and Technology Grant MOST 107-2313-B-001-007-MY2 to YICH, and Academia Sinica Investigator Award. URL of MoST: https://www.most.gov.tw/?l=en URL of Academia Sinica: https://www.sinica.edu.tw/en The funders had no role in study design, data collection and analysis, decision to publish, or preparation of the manuscript.

**Competing interests:** The authors have declared that no competing interests exist.

**Abbreviations:** ANOVA, analysis of variance; CO, *CONSTANS*; Ehd1, *Early heading date 1*; FT, *FLOWERING LOCUS T*; Ghd7, *Grain number, plant height, and heading date 7*; Hd, heading date; Hd1, *Heading date 1*; IGV, Integrative Genomics Viewer; indels, insertions/deletions; IRRI, International Rice Research Institute; LD, long-day; LTRs, long terminal repeats; LOF, loss-of-function; NCBI, National Center for Biotechnology InformationORF, open reading frame; OMAP, *Oryza* Map Alignment Project; QTL, quantitative trait loci; SAM, sequence alignment/map format; SD, short-day; SNP, single nucleotide polymorphism; TC65, Taichung65; VCF, *variant call format*.

Flowering time, also known as heading date (Hd) for rice, is an important trait in the cultivation period [2]. Flowering time is affected by photoperiod (day length) and also temperature [3]. As a facultative SD plant, rice flowering is promoted under SD conditions and delayed under long-day (LD) conditions, with a critical day length of about 13 hours [4, 5]. Several recent reviews provided detailed information on the regulation of rice flowering (e.g., Itoh and Izawa, 2013 [6]; Tsuji et al., 2013 [7]; Shrestha et al, 2014 [8]).

A few key genes have been suggested to affect flowering time, including *Heading date 1* (*Hd1*), which encodes a B-box zinc finger protein and is the ortholog of Arabidopsis *CONSTANS* (*CO*) [9]; *Early heading date 1* (*Ehd1*), which encodes a B-type response regulator with no ortholog in Arabidopsis [10]; *Hd3a*, which encodes a phosphatidylethanolamine-binding protein and is an ortholog of Arabidopsis *FLOWERING LOCUS T* (*FT*) [11]; and *Grain number, plant height, and heading date 7* (*Ghd7*), an important regulator of Hd and yield potential [12].

In the temperate area, including Japan, Korea, northern China and Europe, rice crops grow once a year (i.e., they are germinated/transplanted in early summer and harvested in late fall). However, in subtropical and tropical areas, where the temperature is suitable for plant growth from February to November or even the whole year round, changes in sensitivity to day length played important roles during rice domestication/adaptation. If the cropping season each year may be increased from 1 to 2 or 3, the total seed production would at least double. Today, there are 2 cropping seasons per year in Taiwan and Vietnam and 2 or 3 in the Philippines, Thailand, India, and Pakistan. Thus, these varieties must be insensitive to photoperiod.

Decades ago, cultivated rice varieties were found to have a large variation in flowering time, also known as Hd [5]; that is, rice flowering was controlled by quantitative trait loci (QTL). By using the progeny derived from a single cross between a *japonica* rice variety, Nipponbare, and an *indica* variety, Kasalath, Yano and colleagues identified 15 QTL for *Hd*: *Hd1* to *Hd5* were mapped by QTL analysis of an $F_2$ population [13], and *Hd7*, *Hd8*, and *Hd11* were discovered with $BC_1F_5$ lines [14]. Seven loci, *Hd6*, *Hd9*, *Hd10*, *Hd12-Hd15*, were detected by using advanced backcross generations [15, 16]. The molecular genetic pathway for 2 model plants, Arabidopsis (an LD plant) and rice (an SD plant), has been addressed in many studies and the pathways were recently compared (e.g., Cho et al., 2017 [17]; Hori et al., 2016 [18]; Shrestha et al., 2014 [8]).

Rice *Hd1* was identified as a *CO* ortholog and encodes a zinc-finger type transcriptional activator containing CO, CO-like, and TOC1 (CCT) domains [9]. It has 2 exons, with the CCT domain at the second exon. This CCT domain functions as a nuclear localization signal, and the mutant without the CCT domain in CO showed a defect in protein function [19]. Early rice genetics studies also assigned this locus as *Se1* [20], *fl* [21], *Lm* [22], *K* [23] and *Rs* [24]. All of these alleles were then confirmed to be identical to *Hd1* by linkage analysis (summarized by Kinoshita, 1998 [25]). The functional Hd1 protein is required for suppressing flowering under LD conditions and promoting flowering under SD conditions. *Hd1* regulates *Hd3a* expression [26], the rice ortholog of Arabidopsis *FT*, which encodes a mobile flowering signal [11]. However, *Ehd1*, promoting SD flowering in the loss-of-function (LOF) *hd1*, encodes a B-type response regulator but has no ortholog in the Arabidopsis genome. Variations in Hd1 protein, *Hd3a* promoters, and *Ehd1* expression contribute to the diversity of flowering time [26].

To reveal the selection footprint in *Hd1* in Asian rice cultivars, Fujino et al. (2010) [27] used a 2.9-kb *Hd1* gene region, from the promoter to 3′ UTR, with 60 landraces, including *indica*, *japonica*, and *aus*, for allele network analysis [28]. The authors showed that non-functional alleles were not clearly associated with cultivar groups or locations. Their studies also revealed that multiple introgressions in the *Hd1* gene region had occurred and were important for rice accessions to adapt to different habitats. Furthermore, a set of chromosome segregation

substitution lines was used to elucidate the allele variations of common rice flowering-time QTL [29]. *Hd1* was one of the 8 genes under investigation. The authors also discovered that genomic rearrangements around the *Hd1* region occurred at a relatively early time during rice expansion. They suggested that the functional diversity of these genes was not related strongly to the phylogenetic relationship of subspecies.

In the present study, we analyzed *Hd1* gene sequence changes by using the 3K rice genome sequence database and sequence information for several local rice accessions as well as weedy rice (*O. sativa f. spontanea*) and wild rice. Together with the haplotypes discovered previously, we investigated 9 LOF *hd1* haplotypes. The sequence information was compared with the flowering date data, along with locations of these rice accessions. Two haplotypes were more abundant than others, had been expanded to nearby areas and are used extensively in modern breeding programs. Selection parameters and allele network analyses revealed that these 2 mutations occurred in tropical insular Asia areas early during rice cultivation, then expanded to nearby areas.

## Materials and methods

### Classification of *Hd1* haplotypes and DNA sequence analyses

Three sets of whole-genome sequencing data were downloaded from the public domain: rice 3K project [30], Chinese weedy rice [31] and Asian wild rice [32, 33]. The paired reads were mapped against the Os-Nipponbare-Reference-IRGSP-1.0 database [34, 35]. SAMtools [36] and VCFtools [37] were used to manipulate and transform the sequence alignment/map format (SAM) and variant call format (VCF) of the file. To detect single nucleotide polymorphisms (SNPs) and small indels, we used the command lines in the section "Variant Calling" in "Workflows" of the SAMtools manual without any restriction on depth or mapping quality. The information on SNPs and small indels was recorded in VCF files.

The functional impact of nucleotide variants was analyzed by using the rice genome sequencing data with SnpEff [25]. From the genome annotation, sequence variants were classified according to their location (open reading frame, intron, splice sites, etc.) and predicted functional impact (missense, frame shift, early stop, etc.).

To search for the 1.9-kb deletion in the *Hd1* region in the 3K dataset, we performed a quick screening by using a python script (S1 File). We first identified the corresponding location of the 1.9-kb insertion in Nipponbare using TC65 sequence data, then tried to find any "unreasonable unpaired read" among the Illumina paired-end reads in this region because the distance between each pair should be about 200 to 300 bp. We then verified each candidate by using Integrative Genomics Viewer (IGV, [38–40]). Candidates with similar signature to Taichung65 (TC65) at the site of the 1.9-kb insertion were the accessions carrying the type 19 *hd1* gene.

To search for the 36-bp insertion at the first exon that was not present in the Nipponbare genome but was present in Kasalath [9], we used the insertion position with BAM files and IGV for visualized identification. To accelerate the process, we used the batch function of IGV to produce snapshots of the insertion region. The IGV images for Nipponbare (without the 36-bp insertion) and Kasalath (with the 36-bp insertion) (S1 Fig) were different and were used as a standard to scan all accessions.

To reveal the relationship of 3K landraces carrying type 7 and 13 *hd1* mutations, we used phylogenetic analysis with next-generation sequencing data. S1 Table lists the names, types, origins and sequence information for these lines. The clean reads were mapped to the Nipponbare reference genome (IRGSP v1.0) by using BWA v0.7.13-r1126 mem with default parameters [35, 41]. The mapped results were merged, and data with low mapping quality (q<20)

were removed as BAM files by using Samtools v1.3 [36, 42]. Picard v2.1.1 MarkDuplicates was used to identify and remove duplicate reads in the same DNA fragments (http://picard.sourceforge.net). The Genome Analysis Toolkit v3.5-0-g36282e4 RealignerTargetCreator was used to identify regions around indels, then the Genome Analysis Toolkit IndelRealigner was used for local realignment [43]. Samtools and Bcftools were used for variant calling including SNPs and indels with filter by depth and mapping quality. Genetic distance was calculated with the p-distances model, and a neighbor-joining tree was constructed with 1,000 bootstraps by using PHYLIP v3.695 [44]. MEGA v7 [45] was used to display the phylogenetic tree.

High-quality SNPs were selected from VCF files according to 3 criteria: 1) homozygous, 2) genotype quality >20, and 3) allelic depth > 4x. The VCF files were then converted to the NEXUS multiple alignment format by using vcf2phylip v2.0 [46]. Allelic haplotype networks were then created by using TCS networks v1.13 [28] with the parsimony connection limit as 2 steps and gaps treated as missing data.

### Estimation of diversity of different *Hd1* haplotypes

DNA sequences were aligned by using MUSCLE [47, 48]. The −10- to +10-kb region of *Hd1* genes, corresponding to Nipponbare genome chromosome 6 from 9,326,376 to 9,348,569, was used for the analysis. Statistical analysis involved using DNAsp.v6 [49]. Items analyzed included number of polymorphic (segregating) sites (*S*); total number of mutations (*Eta*); average number of nucleotide differences (*k*); nucleotide diversity ($\pi$ [50]); *theta* (per sequence) from *Eta* $\theta$; and Watterson's estimator of *theta* (per site) from *Eta* $\theta w$ [51]. The neutrality test, Tajima's *D* value [52], was used to test the neutral mutation hypothesis. The *D* value was based on the discrepancy between $\pi$ and $\theta w$. Thus, negative values indicate excess low-frequency polymorphism. These values were calculated after removing missing data and alignment gaps.

## Results

### Many LOF haplotypes are present in rice accessions

Takahashi et al. (2009) [26] identified a high degree of sequence diversity in rice *Hd1* and grouped them into 17 types. The previously recognized Kasalath haplotype [9], with a 2-bp deletion and thus LOF, was classified as type 13. In addition, the authors discovered 4 new mutations leading to LOF *hd1*, including 3 deletions and 1 SNP. Both types 2 and 3 had a 1-bp deletion, type 7 a 4-bp deletion, and type 12 an SNP inducing an early stop.

In the first work for *Hd1* gene cloning, Yano et al. (2000) [9] reported 2 haplotypes that induced LOF *hd1*: the 2-bp deletion in Kasalath (*hd1*) and the 43-bp deletion (*se1*) in HS66, a mutant of a Japanese landrace Ginbozu located on the short arm of chromosome 6. This 43-bp deletion was not grouped previously, and here we classified it as type 18. Doi et al. (2004) [10] worked on *Ehd1* and *Hd1* genes of a photoperiod-insensitive variety TC65. The TC65 *hd1* haplotype has a 1.9-kb insertion at exon 2. However, this TC65 haplotype was not mentioned in Takahashi's nomenclature [26]. In the current study, we classified it as type 19. In our previous study, we demonstrated that this 1.9-kb insertion was introgressed from 2 Taiwan landraces, Muteka and Nakabo, to the early breeding generation of TC65 [53].

In the current study, we discovered 3 new LOF *hd1* types from the 3K [30] and *Oryza* Map Alignment Project (OMAP) [32] rice genome sequencing information: type 20 with an A-to-T SNP at position 9338273 that led to an early stop at Arg368; type 21 with another SNP (C to T) at position 9337150 for an early stop at Gln206; and type 22 with a C-to-A change at position 9337112 for an early stop at Ser193.

These types could be grouped into 22 types, with 10 mutations inducing loss of Hd1 protein function: types 2, 3, 7, 12, 13, 18, 19, 20, 21 and 22. S2 Table shows the accessions, types and

**Table 1. Haplotypes of loss-of-function *Hd1* genes in the 3K rice dataset.**

| NB location | 9336853 | 9337102 | 9337112 | 9337150 | 9337242 | 9338004 | 9338032 | 9338220 | 9338243 | 9338273 |
|---|---|---|---|---|---|---|---|---|---|---|
| Types of aa changes | FS* | FS | STOP | STOP | FS | FS | FS | FS | STOP | STOP |
| Nipponbare seq † | C | G | C | C | - | TTT | - | AAGA | C | A |
| Type 2 | | 1-bp del | | | | | | | | |
| Type 3 | 1-bp del | | | | | | | | | |
| Type 7 | | | | | | | | 4-bp del | | |
| Type 12 | | | | | | | | | T | |
| Type 13 | | | | | | 2-bp del | | | | |
| Type 18 | | | | | 43-bp del | | | | | |
| Type 19 | | | | | | | 1.9 kb | | | |
| Type 20 | | | | | | | | | | T |
| Type 21 | | | | T | | | | | | |
| Type 22 | | | A | | | | | | | |

Nipponbare (NB) position, nucleotide sequence changes and types of protein changes are indicated.

*FS: frame shift.

†: The position on Os-Nipponbare-Reference-IRGSP-1.0.

countries for all 10 LOF haplotypes in the 3K resources. The mutations included 3 categories: 1) a small insertion/deletion (indel) leading to a frameshift: types 2, 3, 7, 13 and 18; 2) an SNP leading to an early stop: types 12, 20, 21 and 22; and 3) a large (1.9-kb) insertion leading to a frameshift: type 19. Previously when types 1 to 17 were defined, the genome sequences of land-race Ginbouzu were used [26]. Here we aligned changes to the Nipponbare reference sequence, and Table 1 shows the haplotypes of the LOF *hd1* gene in the 3K rice database as well as its sequence localization and changes in amino acid residues.

The Hd information for rice grown at the International Rice Research Institute (IRRI) campus, along with abundant phenotype data for about two-thirds of the 3K accessions are available in the IRRI SNP-SEEK phenotype database (https://snp-seek.irri.org/) [54]. Type 18 belongs to a mutation in the mutagen-induced mutant population of Ginbouzu, and type 22 occurs only in *O. punctata*, an African wild rice, so no phenotype data are available for these 2 types in SNP-SEEK. The Hd information was downloaded from the database and grouped according to 8 LOF haplotypes. The ANOVA results indicated that the Hd for these 8 groups significantly differed from that of the wild type (S3 Table), which did not contain any of these 8 haplotypes. Because only one accession of type 2 had Hd information, the Hd behavior was analyzed by *t*-test for only 7 haplotypes. Fig 1 illustrates the Hd phenotype for these haplotypes and the wild type (i.e., the accessions without any LOF haplotypes), with the detailed data in S4 Table. The day length for Los Baños, on the IRRI main campus, ranges from 11 hr 17 min to 12 hr 57 min year-round; the critical day length of rice is about 12.5 hr. Wildtype accessions had an average Hd of 106 days, and Hd for the other types ranged from 77 to 93 days (S4 Table). Because all haplotypes tested flowered before the wild type did, we confirmed that the accessions with these 7 types were indeed not sensitive to day length.

### Many accessions have types 7 or 13 LOF *hd1*, as revealed by the 3K genome data

A total of 408 accessions in the 3K genome had a type 7 deletion and thus LOF *hd1*; 99% are *indica* rice. In total, 51 accessions belong to traditional accessions: 16 were from Indonesia, 5 the Philippines, 4 Bangladesh, 4 Laos, and 3 Vietnam. All are *indica* rice (Table 2). A total of

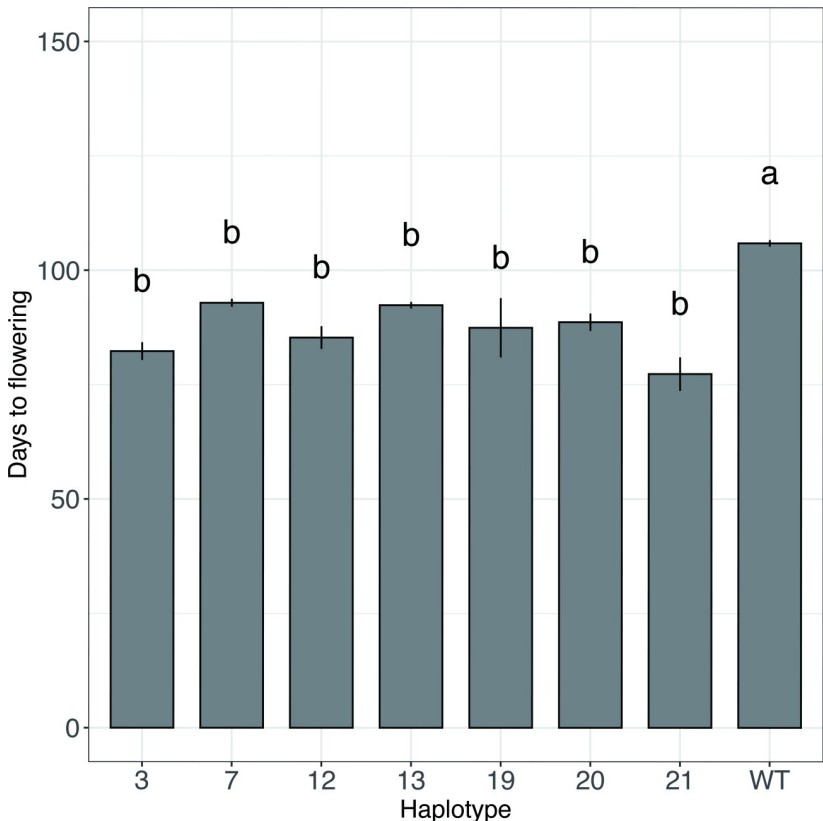

**Fig 1. Flowering date of the 7 loss-of-function haplotypes of *Hd1* gene and the wild type.** Plants were grown at the International Rice Research Institute and the flowering date information was downloaded from the SNP-SEEK website (https://snp-seek.irri.org/). Sample number and detailed statistical data are in S4 Table. WT, wild type. Different letters above bars indicate significant difference at P<0.05.

772 accessions had the type 13 *hd1* allele (i.e., the Kasalath type); 291 belong to traditional landraces: 35 are from Bangladesh, most *aus* rice; 99 from Indonesia, 61 *japonica*, and 34 *indica*; and 4 are intermediate types; 62 from the Philippines, 47 *japonica* and 15 *indica*; 17 from Malaysia, 16 *japonica* and 1 *indica*; 16 from India, all *indica* or *aus*; 13 from Pakistan, all *aus*; 7 from Sri Lanka, 6 *indica* and 1 *japonica*; and 5 from Thailand, 3 *indica* and 2 *japonica*. All are tropical areas. In addition, many accessions belong to tropical *japonica* and are from insular countries such as Indonesia, the Philippines and Malaysia (Table 2, S2 and S5 Tables). To summarize, about 800 modern rice varieties had type 7 or 13 LOF *hd1*. Thus, types 7 and 13 are used intensively in modern breeding programs in many countries.

## The 1.9-kb insertion in type 19 is a small retrotransposon

The insertion in type 19 is 1901 bp long at exon 2. Dot matrix analysis of this sequence revealed long terminal repeats (LTRs) at both sides, 448 bp in length, and with exact identity between the 2 LTRs. In addition, the central part of this fragment contains a partial pol protein sequence. Hence, this is the smallest retrotransposon in the rice genome. The cultivated rice genome, including both *japonica* and *indica* rice, contains 2 similar retrotransposons, one at chromosome 5 and another at 8, with length 1895 and 1893 bp, respectively. In the Nippon-bare genome, the retrotransposon in chromosome 5 is 446 and 452 bp for the right and left LTR, respectively, with 97% identity of these 2 LTRs. The retrotransposon in chromosome 8 is 445 and 450 bp for the left and right LTR, respectively, with identity 97%.

**Table 2. Traditional accessions of 8 haplotypes in each country.**

| | Total | Traditional | Allele group | | | | | | | |
|---|---|---|---|---|---|---|---|---|---|---|
| | | | 2 | 3 | 7 | 12 | 13 | 19 | 20 | 21 |
| Bangladesh | 186 | 104 | – | -- | 4 | -- | 35 | -- | -- | -- |
| Bhutan | 19 | 19 | -- | -- | 2 | -- | 2 | -- | -- | -- |
| Cambodia | 59 | 45 | -- | -- | -- | -- | 1 | 1 | -- | -- |
| China* | 481 | | -- | (11) | (120) | -- | (50) | (16) | -- | -- |
| India | 435 | 73 | -- | 2 | 3 | -- | 16 | -- | 9 | 3 |
| Indonesia | 248 | 203 | -- | -- | 16 | -- | 99 | -- | -- | -- |
| Japan | 55 | 11 | -- | -- | -- | -- | -- | -- | -- | -- |
| Korea | 34 | 19 | -- | -- | 1 | -- | -- | -- | -- | -- |
| Laos | 126 | 84 | -- | -- | 4 | 10 | -- | -- | -- | -- |
| Madagascar | 66 | 16 | -- | -- | -- | -- | 8 | -- | -- | -- |
| Malaysia | 75 | 55 | -- | -- | 2 | -- | 17 | -- | -- | -- |
| Myanmar | 75 | 21 | -- | -- | 1 | -- | -- | -- | -- | -- |
| Nepal | 44 | 27 | -- | -- | -- | -- | 3 | 1 | -- | -- |
| Pakistan | 34 | 24 | -- | -- | 1 | -- | 13 | -- | -- | -- |
| Philippines | 229 | 97 | -- | -- | 5 | -- | 62 | -- | -- | -- |
| Sri Lanka | 54 | 40 | -- | -- | 1 | -- | 7 | -- | -- | -- |
| Taiwan | 30 | 23 | -- | 5 | 4 | -- | 3 | 11 | -- | -- |
| Thailand | 147 | 103 | -- | -- | 1 | -- | 5 | -- | -- | -- |
| Viet Nam | 55 | 28 | 2 | -- | 3 | -- | 1 | -- | -- | -- |
| Total | 2457 | 990 | 2 | 7 | 48 | 10 | 272 | 13 | 9 | 3 |

Most accessions are 3K data, and we added some accessions collected from Taiwan aboriginal villages [53].

*Because of no information on traditional/modern types for Chinese accessions, the total numbers are shown in brackets.

## Other haplotypes occurred in local regions

Only 2 accessions in the 3K collection had type 2 *hd1*. Both are *indica* rice and were from Vietnam. A total of 21 accessions had type 3, all *indica*; 11 are from China and 5 from Taiwan. These 5 accessions were brought to Taiwan from southern China in the late Ming Dynasty about 400 years ago and thus must be old landraces. A total of 28 accessions had type 12; 10 are traditional landraces. These landraces were from Laos, and all are tropical *japonica*. In total, 31 accessions had type 19 *hd1* and most are modern varieties, including TC65, Taichung 179 and Taichung 188. According to our previous study of the *Hd1* gene in Taiwan aboriginal rice accessions using genomic PCR analysis, 11 had this haplotype [53]. A total of 16 accessions had type 20, all from India and all *indica* or *aus*. Only 3 accessions had type 21, all from India and all *aus* (S2 Table).

## LOF haplotypes also found in Chinese weedy rice accessions and an African wild rice accession

In addition to many landraces and modern varieties, much effort has been extended to resequencing the many weedy rice accessions (Qiu et al., 2017 [55] and Sun et al., 2019 [56] for Chinese ones; Li et al., 2017 [57] for American ones) and wild rice accessions [32, 33, 58]. We downloaded the sequence information for 155 Chinese accessions from the US National Center for Biotechnology Information (NCBI) for *Hd1* haplotype analysis. The accessions collected from Liaoning (northeastern part) and Ningxia (northwestern part) are *japonica* and those from Jiangsu (central part) and Guangdong (southern part) are *indica* [55]. In all, 25 of the 27

Guangdong accessions had type 7, one accession did not have any known LOF haplotype and one accession had heterozygous type 7. However, all 39 Jiangsu accessions and 30 Ningxia accessions had only functional *Hd1*. For the 59 Liaoning accessions, 2 had type 7 and another 2 type 13; the remaining 55 accessions had functional *Hd1*. The detailed analysis is in S6 Table.

Rice production in the Guangdong region involves 2 cropping seasons, with only one cropping season in the other 3 regions. Sequence analysis also indicated only Guangdong with the LOF *hd1* gene but not the other 3 regions. Thus, haplotype analysis indicated that flowering habitat of these weedy rice accessions had been adapted to the nearby rice production, and only those in southern China had an LOF *hd1* allele.

We also performed detailed analysis of *Hd1* haplotypes with the resequencing data for wild rice accessions downloaded from NCBI. Only accessions with at least 10X genome redundancy were used, including those from 1) the *O. rufipogon* pangenome project: W0123, W0141, W0170, W1687, W1698, W1739, W1754, W1777, W1943, W1979, W2012, W3078, and W3095 [33]; and 2) the *Oryza* map alignment project: *O. barthii*, *O. glumaepatula*, *O. meidionalis*, *O. nivara*, *O. rufipogon* and *O. punctata* [32]. A new LOF *hd1* haplotype was found in *O. punctata* (S2 Table). This type 22 is an SNP of C to A at position 9337112, leading to an early stop of Hd1 protein.

## Phylogenetic and haplotype network analyses of types 7 and 13

The distribution of types 7 and 13 *hd1* are relatively wide and the accession numbers high, including the landraces and modern varieties; that is, about 40% of the 3K project had either of these 2 mutations, which indicates their importance in primitive and modern rice production in many areas. Thus, we further analyzed these 2 types by using phylogenetic relationship and allele network analyses. We used the *Hd1* gene region, a 3.8-kb fragment from −1-kb to +1-kb, for elucidating the relationship. These landraces include *japonica*, *indica*, *aus* and *aromatic* types from 4 regions: 1) insular Southeastern Asia areas (i.e., Indonesia, the Philippines, and Malaysia); 2) Indochina areas (i.e., Cambodia, Laos, Thailand, Vietnam, and Myanmar); 3) Indian subcontinent (i.e., Bhutan, Nepal, India, Sri Lanka, Bangladesh, and Pakistan); and 4) Madagascar. Fifteen Asian wild rice accessions were also used in the analysis, including *O. rufipogon* and *O. nivara* sequences from the OMAP project [32] and the 13 wild rice pan-genome sequences [33]. The accessions belonged to OrI, OrII, OrIIIa or OrIIIb and were collected from Indonesia, Malaysia, India, Thailand and China [32, 33]. The information on accessions, types and collected regions is in S1 Table.

Nucleotide changes in the *Hd1* gene region of the 40 landraces with type 7, the 262 landraces with type 13, and the 15 wild rice accessions are in S7 and S8 Tables. These 302 landraces feature 41-nt changes, 22 in the open reading frame (ORF). The 40 landraces with type 7 are mainly *indica* rice and may be grouped into 5 types, 34 with the first 2 types. However, accessions with type 13 are more complicated: 262 landraces were grouped into 156 types, 28 with the first 2 types. In addition, they belong to tropical *japonica*, temperate *japonica*, *indica*, *aus* and *aromatic*. There are more nucleotide changes in the 15 Asian wild rice accessions than landraces, each accession with a unique type (S8 Table). This SNP information was used to construct a phylogenetic tree and network.

The phylogenetic tree indicated that the landraces with type 7 are clustered together and are distinct from the 15 wild rice accessions (Fig 2A). However, the relationships of landraces with type 13 are rather complicated. These landraces may be grouped into 5 groups: group I is closest to the wild rice group, which indicates that they belong to the most ancient changes. More than 80% are from insular Southeastern Asia areas, so the early type 13 might occur in the region. Group V were mainly *aus* from the Indian subcontinent (highlighted in blue), and the

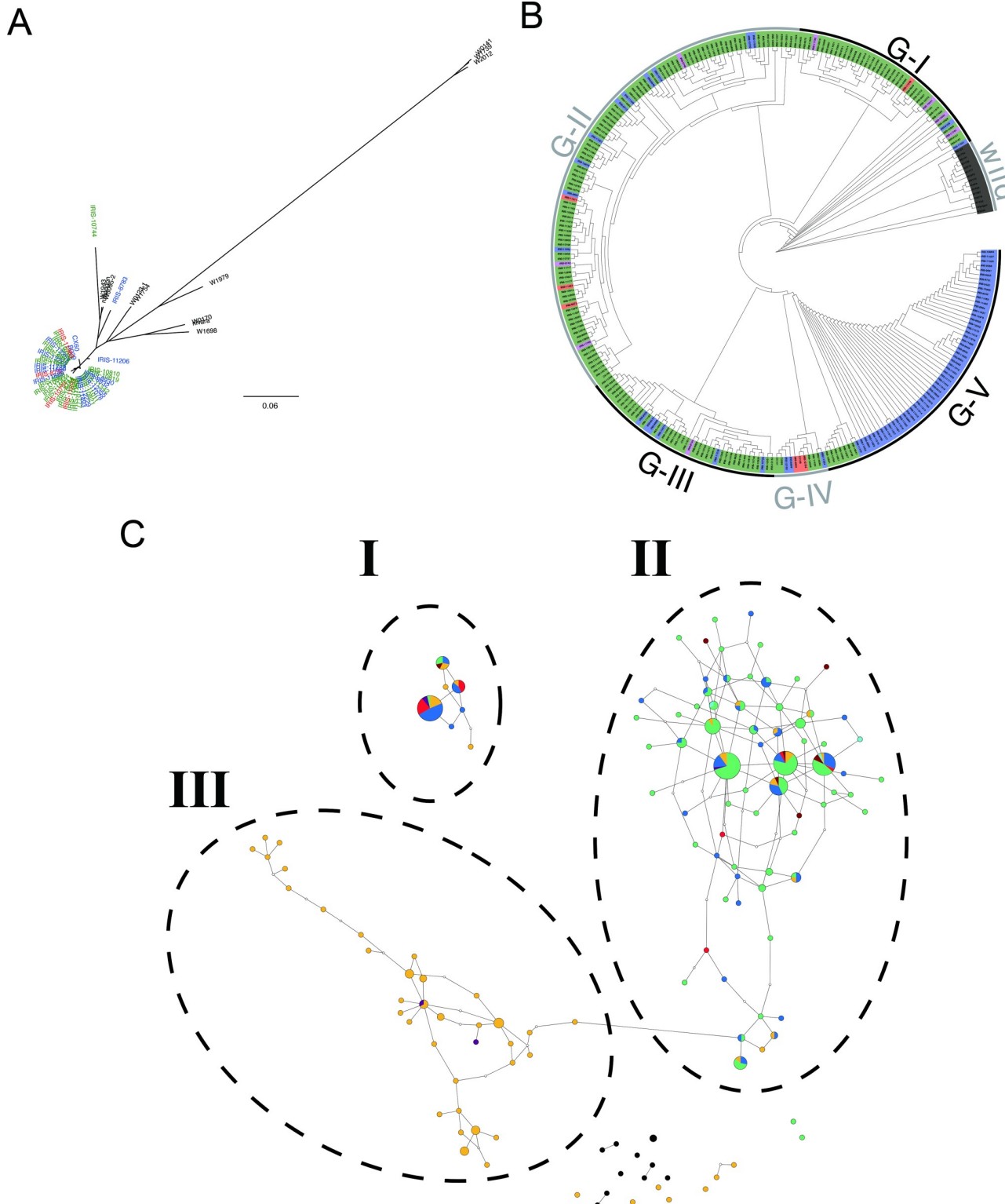

**Fig 2. Phylogenic and network analysis of types 7 and 13 *Hd1* gene.** Nucleotide changes in the −1000 to + 1000-bp *Hd1* region among 3K Asian landraces carrying types 7 and 13 loss-of-function mutation haplotypes as well as 15 Asian wild rice accessions were used. Accession information is in S1 Table. **A.** Phylogenetic tree of 40 3K accessions carrying type 7 *Hd1* mutation and 15 wild rice accessions. The tree was constructed with SNP data by using PHYLIP [44]. Green indicates insular Southeastern Asia areas, red Indochina areas, blue Indian subcontinent, magenta Madagascar and black wild rice species. The bootstrap values determined with 1,000 samples are shown. **B.** Phylogenetic tree of 262 3K accessions carrying type 13 *Hd1* mutation

and 15 wild rice accessions. Groups I to V are indicated. Other notes are the same as **A. C.** Allele network of types 7 and 13 as well as the wild rice accessions was constructed with the TCS program v1.13 [28]. Black circles indicate wild rice accessions, green and blue highlight *japonica* and *indica* types from insular Southeastern Asia areas, cyan and red highlight *japonica* and *indica* for Indochina areas, purple and orange highlight *japonica* and *indica* for Indian subcontinent, and brown highlights *indica* for Madagascar. Sizes of circles are proportion to the haplotype frequencies. Lines indicate mutation steps.

clade originated from 7 accessions (highlighted in green at group V) from insular Southeastern Asia areas, with 6 tropical *japonica* and 1 *indica* accessions (Fig 2B).

The allele network revealed 3 groups for type 7 and 13 *hd1*: group I were all accessions with type 7; groups II and III both had accessions with type 13 (Fig 2C). Most of the accessions in group I were from insular Southeastern Asia areas, with a few from Indochina and the Indian subcontinent. Group II accessions were rather complicated, most being tropical *japonica* accessions from insular Southeastern Asia areas. The minor ones involved *indica* in the same area as well as *japonica* and *indica* in the Indochina area. Group III was clearly a separate lineage diverged from a small group of *japonica/indica* accessions from insular Southeastern Asia areas. All these accessions were *aus* collected from the Indian subcontinent and all except 2 had identical SNPs at the intron, the second ORF, and 3′ region (S7 Table). The first several "dots" of group III connecting to group II were collected from Bangladesh, fit well with the geographic distribution and indicated the footprint of early human movement. About 70% of group III accessions were collected from Bangladesh. This coalescent approach clearly suggested that type 13 occurred at insular areas in Southeastern Asia, followed by introgression and expansion (i.e., brought by human beings) to the Indochina area and Indian subcontinent. In addition, both type 7 and 13 did not have a close relationship with the Asian wild rice accessions tested. Hence, allele network analysis clearly shows the mutations, introgression and human activity of these accessions during rice cultivation/adaptation.

During the work on the allele network analysis, we found that 3 wild rice accessions, W3078, W1687 and W1777, must have had introgressions in the 3.8-kb *Hd1* gene region. The W3078 genome is completely identical to that of Nipponbare, and the other 2 have 2 and 4 SNPs, respectively. Thus, only 12 wild rice accessions were used for further allele network analysis. This finding also revealed that some of the wild rice accessions had introgression fragment(s) from cultivars.

## Discussion

Together with *Ehd1*, *Hd1* was suggested to control rice panicle development and showed their effect in crop yield [59]. Thus, *Hd1* could have dual effect on rice production (i.e., involved in rice development and controlling flowering time). Insensitivity to photoperiod in tropical and subtropical regions has been beneficial to rice production because of 1) increased crop yield per year and 2) escape/avoidance from stresses caused by seasonable typhoons, monsoons, or drought. Thus, insensitivity was one of the targets to select for heading gene mutations during the long rice cultivation history. In this study, we analyzed the LOF *hd1* mutations that occurred in rice in different regions of Asia by using whole-genome sequencing data for thousands of accessions.

### *Hd1* was suggested to be the main allele associated with adaptation of rice plants to tropical regions

Rice was domesticated about 8,000 to 10,000 years ago in the Yangtze River region of China, which today still has one cropping season. Domesticated rice had spread southward thousands of years ago, and one of the new traits beneficial to crop yield would be loss of sensitivity to

photoperiod. If rice could be cultivated for 2 or 3 seasons each year, production would at least double.

Takahashi and Shimamoto [2] performed a cDNA sequencing analysis of *Hd1* and *Hd3a* genes in leaf tissues at tillering stage under an SD condition from several rice cultivars and 38 wild rice accessions. Cultivated and wild rice accessions showed no nucleotide changes in *Hd3a*. Also, the 38 wild rice accessions showed no change affecting *Hd1* function, but several changes appeared in cultivated rice. Thus, the authors suggested that *Hd1* was a possible target of selection to generate different flowering-time responses in different regions. Kim et al. [60] selected about 60 diverse rice accessions to analyze the Hd trait of various types, including 4 *aus* accessions, 20 *indica* accessions, 4 tropical *japonica* accessions, 17 temperate *japonica* accessions, and 12 tropical adapted temperate *japonica* accessions. The authors analyzed 7 major flowering genes, including *Hd1* [9], *OsPPR37* [61], *DTH8* [62], *Ghd7* [12], *Ehd1* [10], *RFT1* [63] and *Hd3a* [11] as well as heading behavior under 3 different field conditions in temperate and tropical regions. They concluded that only the accessions from tropical/subtropical regions preferentially had the non-functional alleles of *hd1* but not the other flowering genes tested. Therefore, in the present study, we focused on the *Hd1* haplotype.

### Several *Hd1* haplotypes were limited to small regions and some were widely spread

Table 2 summarizes the traditional accessions for each haplotype in rice-growing countries of Asia. As indicated in Results, types 2, 3, 12, 19, 20 and 21 were limited to small regions. Thus, mutations of these 6 LOF haplotypes must have occurred recently during adaptation, and all were in accessions in tropical/subtropical regions. Most were *indica* or *aus* rice, with only types 12 and 19 as tropical *japonica*. Fig 3 illustrates the sample size and frequency of traditional landraces collected for each haplotype in rice-growing countries in Asia. S2 Fig illustrates the distribution of subtypes in each of these countries to show ratios of *aus*, *indica*, temperate *japonica*, tropical *japonica*, *aromatic*, and admixture. Fig 3 clearly shows that no or only a few number or types of LOF *Hd1* haplotypes were found in high-altitude and -latitude regions such as Bhutan, Nepal, Korea and Japan. However, most of the accessions from tropical/subtropical regions had the non-functional *Hd1* alleles, with multiple haplotypes in the same region.

For the 1.9-kb small retrotransposon integrated into *Hd1* of type 19, because the length and sequences of both LTRs are identical, this transposition into chromosome 6 might have occurred about 2,000 years ago. Two similar retrotransposons at chromosome 5 and 8 had different lengths for the LTR pair, and the sequences were not identical (97% for each), so the transposition into these 2 locations must have occurred a long time ago. In addition, these 2 small retrotransposons were located in chromosomes 5 and 8 in *indica* rice accessions, which further confirmed that they were present in the rice genome before the split of *japonica* and *indica* rice. In the Asian AA-genome wild rice, *O. nivara* has the chromosome 5 copy only, and *O. rufipogon* has both chromosome 5 and 8 copies.

Many traditional landraces were relatively limited to small areas, but this was not the case for accessions with types 7 and 13 (Table 2). S5 Table shows accession numbers with type 7 and 13 *Hd1* alleles in the traditional landraces of each country, including information on their subspecies. Most of the type 7 accessions were *indica* rice, with more *japonica* rice for type 13 accessions, especially in Indonesia and the Philippines. Both haplotypes are still used for many modern *indica* varieties worldwide, such as most IR series inbred lines. The phylogenetic and allele network analyses suggested that these two mutations might have occurred first in insular southeastern Asia areas and then introgressed to many local landraces and moved to other regions.

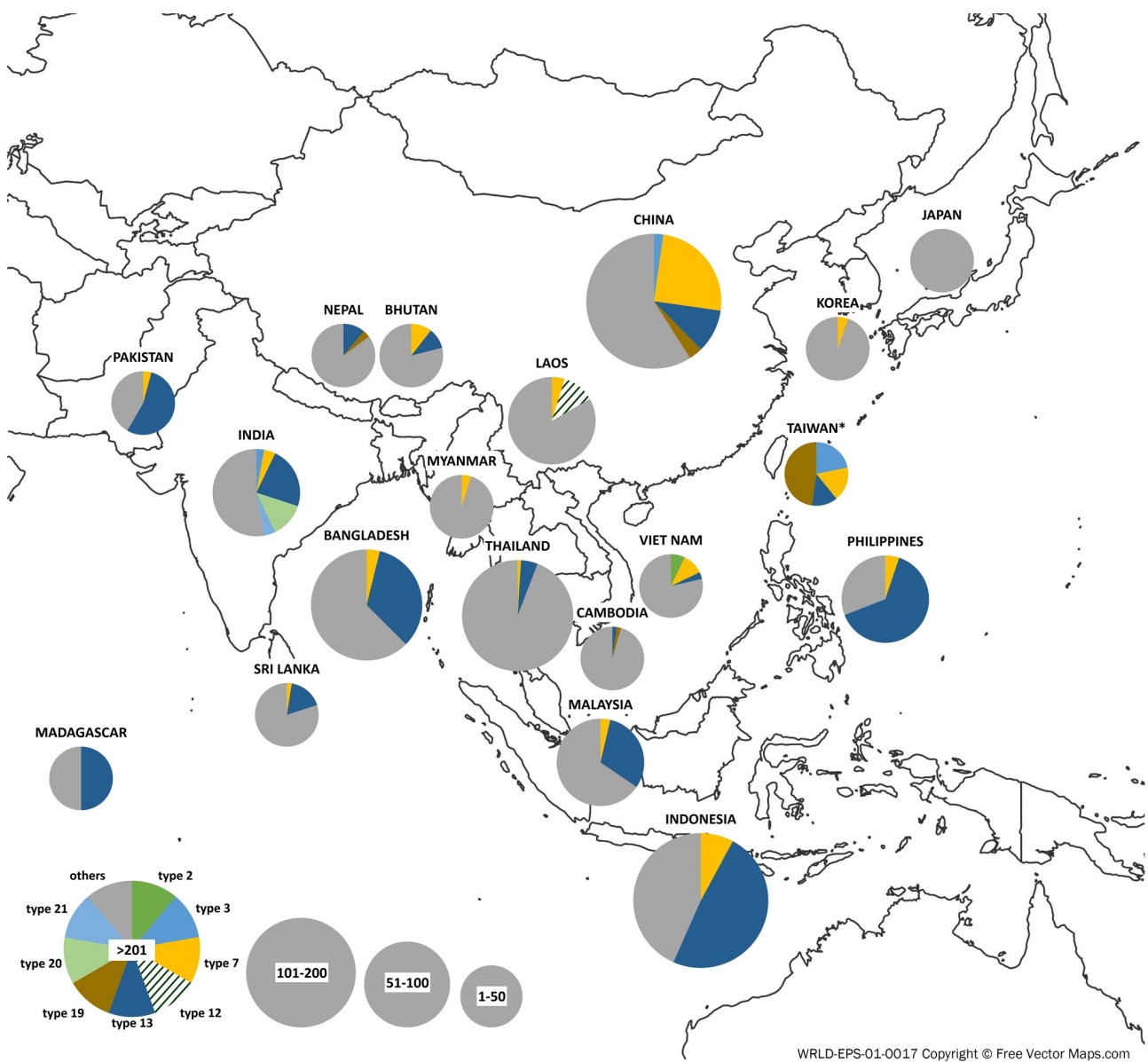

**Fig 3. Geographic distribution of functional and non-functional *Hd1* alleles.** The number in each pie indicates the number of cultivars belonging to each *Hd1* allelic group. Colors indicate different haplotypes and the pie size indicates the sample size. Only the traditional landraces in the 3K dataset are illustrated. Type 2, lime green; Type 3, sky blue; Type 7, yellow; Type 12, green stripe; Type 13, navy; Type 19, brown; Type 20: yellowish green; Type 21, light blue; other (without loss-of-function mutation), gray. The Asia map was downloaded from https://freevectormaps.com/world-maps/WRLD-EPS-01-0017.

## Some weedy rice or wild rice accessions also contained LOF *hd1*

Weedy rice (*O. sativa f. spontanea*), also called "red rice", has been considered a conspecific weed of cultivated rice. Most accessions are shattering, with a long awn, brown pericarp, and dark seed hull. Several origins for weedy rice have been suggested, including gene flow resulting from natural hybridization[64], evolving directly from domesticated lineages[65], introgression with local cultivars coupled with selection that maintained weedy identity[66], influenced by proximity to reproductively compatible wild and domesticated populations[67], originated from *indica-japonica* hybridization [31], hybridization of modern *indica*/*indica* or

*japonica*/*japonica* [68], or evolved from maternal hybrid rice derivatives [69]. As mentioned in the Results section, data mining of the Chinese weedy rice accessions illustrates that type 7 LOF *hd1* occurred in 96% of the accessions collected in Guangdong, a sub-tropical region. However, most of the accessions collected in another 3 locations (all temperate regions) had functional *Hd1*. Because of the ecological meaning of the weedy rice, type 7 *hd1* must have existed in southern China, including Guangdong, for a long time.

With the detailed studies of several wild rice accessions, we also found 1 LOF haplotype in *O. punctata*, a BB genome African wild rice. Therefore, sequence changes leading to LOF *hd1* occurred not only in cultivated rice but also in its wild relatives.

### LOF *hd1* mutations occurred *in agro*

Several independent origins of LOF *hd1* mutations led us to raise interesting questions about the adaptation of the *Hd1*-negative phenotype. One novel mutation, type 22, existed only in an African wild rice accession, and none of the other haplotypes preexisted in the wild rice accessions tested. Thus, most mutations, except type 22, did not originate in wild species but occurred *in agro*.

We then checked for the existence of positive selection in the LOF *hd1* mutation. We calculated selection parameters, including $\pi$ [50], $\theta w$ [51], as well as Tajima's *D* [52], to test the neutral mutation hypothesis. *Hd1* gene and the upstream 10-kb and downstream 10-kb regions were used. A significant negative *D* value indicates strong selection. We analyzed 5 haplotypes in 22 regions. Data are summarized in Table 3 and the complete data are in S9 Table. Type 7 was positively selected in Indonesia but not in Bangladesh, India, Laos, Malaysia, the Philippines or Taiwan. Thus, this mutation occurred in the *indica* rice accession in Indonesia, similar to the allele network analysis. Type 12 was present only in Laos and it was positively selected. For type 13, we found highly significant selection ($P < 0.001$) in landraces of Indonesia and the Philippines, especially in *japonica* rice. The selection was also significant ($P < 0.05$) in Malaysia *japonica* rice. However, the selection was not obvious in Bangladesh or India, etc. These results again verified the allele network studies. Type 19 mutation was mainly found in the aboriginal villages of Taiwan [53] and the parameters shown in Table 3 also indicated positive selection in these lines.

We detected significant Tajima's D values in several types/locations, as shown described above. The negative D values indicate either high selection pressure and thus reducing

**Table 3. Selection pressure analysis of four *Hd1* types in several countries.**

| Haplotype | Country | Subspecies | No. of accessions | π | θw | Tajima's D | Significance of D |
|---|---|---|---|---|---|---|---|
| Type 7 | Bangladesh | *Indica* | 4 | 0.00065 | 0.00066 | -0.15923 | P >0.10 |
| Type 7 | Indonesia | *Indica* and *japonica* | 16 | 0.00138 | 0.0027 | -2.12711** | P <0.01 |
| Type 7 | Laos | *Indica* | 4 | 0.00044 | 0.00044 | -0.15777 | P >0.10 |
| Type 12 | Laos | *Japonica* | 10 | 0.00115 | 0.00179 | -1.75625* | P <0.05 |
| Type 13 | Bangladesh | *Indica* | 35 | 0.00294 | 0.00603 | -1.95450* | P <0.05 |
| Type 13 | India | *Indica* | 16 | 0.00320 | 0.00267 | 0.86860 | P >0.10 |
| Type 13 | Indonesia | *Indica* and *japonica* | 99 | 0.00120 | 0.00547 | -2.65204** | P <0.001 |
| Type 13 | Indonesia | *Japonica* | 61 | 0.00137 | 0.00534 | -2.65105** | P <0.001 |
| Type 13 | Philippines | *Indica* and *japonica* | 62 | 0.00125 | 0.00432 | -2.52823** | P <0.001 |
| Type 13 | Philippines | *Japonica* | 46 | 0.00137 | 0.00432 | -2.50592** | P <0.001 |
| Type 19 | Taiwan | *Japonica* | 11 | 0.00205 | 0.00328 | -1.81600* | P<0.05 |

*Hd1* gene region and nearby ± 10 kb of traditional landraces were used for calculation.

sequence variation or a recent population expansion. A total of 1181 accessions, including landraces and modern varieties, in the 3K resource have types 7 or 13, which indicates that the traits are important locally and globally. Therefore, we suggest that strong selections were applied to these LOF haplotypes (types 7, 12, 13, 19). In total, 203 Indonesian landraces were used in 3K data. However, according to the Genesys website (www.genesys-pgr.org), 6899 Indonesian landraces are in the IRRI collection. That is, we are only studying a small proportion of the total population in Indonesia and in many other countries.

## A 36-bp deletion in the first ORF of Nipponbare genome

While working on the map-based cloning of *Hd1* gene decades ago, Yano et al. pointed out that in addition to the 2-bp deletion in the second ORF in Kasalath that caused LOF, there was a 36-bp insertion in the first ORF of Kasalath as compared with Nipponbare [9]. However, this 36-bp fragment was present in the *CO* gene. Thus, Nipponbare contained an in-frame 12 amino acid deletion. The residues in Arabidopsis are LARRHQRVPILP and in Kasalath are LARRHQRVPVAP (i.e., identical in 10 amino acids and similar in 2). This 36-bp fragment is rich in GC and may form a stem-loop structure with the stable free energy of -2.06 kcal mol$^{-1}$ (S3 Fig). Here we screened the presence of this 36-bp fragment in the 3K landraces, weedy rice and wild rice accessions. Among about 2000 accessions screened, most had this 36-bp fragment (S10 Table). That is, all wild rice and *indica* accessions had this short insertion. Few temperate *japonica* accessions are the same as Nipponbare, including 10 from Korea, 6 from Japan, 4 from China and 1 from Taiwan. For the 4 groups of Chinese weedy rice [55], those from Ningxia (temperate *japonica*), Jiangsu (*indica*), and Guangdong (*indica*) also had the 36-bp insertion. Only one-third of Liaoning accessions (temperate *japonica*) did not have the insertion, which might be due to the geographical closeness to Korea. This short deletion might have occurred in northeastern Asia and spread to nearby regions, without changing the flowering behavior.

## Introgressions and human activities played important roles for rice diversification/adaptation as revealed by the *Hd1* gene region

Previous studies using sequencing information of the *Hd1* gene region of 60 landraces [27] or 429 CSSL materials [29] illustrated that multiple introgressions and thus dynamic rearrangements occurred around the *Hd1* gene. In the present study, we showed that the type 7 mutation occurred in *indica* rice in insular Southeastern Asia areas. The mutation was introgressed into 3 *japonica* rice and other *indica* accessions in the same region. The *indica* types were also moved to the Indochina region through human activities. However, the type 13 mutation occurred in tropical *japonica* rice in insular Southeastern Asia areas and some were introgressed into local *indica* types. Some of these *indica* accessions were expanded to Indochina by human activities, most if not all being *indica* type. A few accessions with type 13 mutation were also brought from the islands to Bangladesh and introgressed to the local *aus* accessions. They were now *aus* lines carrying type 13 mutation in the Indian subcontinent. Thus, there should have been frequent transportation among the islands of insular Southeastern Asia areas and also from islands to Indochina as well as the Indian subcontinent hundreds or thousands of years ago. By studying *Hd1* gene sequences, we illustrated that both introgression and human activities played important roles in rice diversification and adaptation.

## Conclusions

In the current study, we chose landraces carrying LOF *hd1* mutations for detailed analysis. We demonstrated that many LOF *hd1* alleles existed in accessions in subtropical and tropical Asia rice-growing areas. Some of these haplotypes were present locally, whereas 2, types 7 and 13,

originated from insular Southeastern Asia areas. They were introgressed to other accessions, then expanded with human migration. These 2 types were spread to many regions and are now used in most of the modern varieties in many countries in different continents.

## Supporting information

**S1 File. Python script used for screening of 1.9-kb insertion in the 3K data.**
(TXT)

**S2 File.**
(RAR)

**S1 Table. The accessions used for the phylogenetic and allele network analysis.** The accessions and collected regions of the landraces and wild rice are listed.
(XLSX)

**S2 Table. The accession, types and collected nations and resource of loss-of-function *Hd1* gene in the 3K dataset.**
(XLSX)

**S3 Table. ANOVA for flowering dates of *Hd1* haplotypes and the wild type.**
(DOCX)

**S4 Table. Summary of statistical analysis of flowering dates of *Hd1* haplotypes and wild type.**
(DOCX)

**S5 Table. Distribution of *japonica* and *indica* rice in traditional accessions from different countries.** Only types 7 and 13 are illustrated. Most are from 3K data and we added some accessions collected from Taiwan aboriginal villages.
(DOCX)

**S6 Table. Accessions, types, collected regions and source for 10 loss-of-function haplotypes of *Hd1* in weedy rice and wild rice.**
(XLSX)

**S7 Table. Nucleotide changes in the *hd1* gene region for landraces carrying types 7 and 13 loss-of-function mutation.** The 3.8-kb fragments from −1000 to + 1000-bp of *Hd1* were used. Position indicates the corresponding location of chromosome 1 in Nipponbare (IRGSP v1.0). Dot indicates the same as Nipponbare reference sequences, "-" indicates deletion in that position. The 3K code, accession names, subspecies, origin, haplotype group and numbers and sequence changes are listed.
(XLSX)

**S8 Table. Nucleotide changes in the *Hd1* gene region for 15 Asian wild rice accessions.** The 3.8-kb fragments range from −1000 to + 1000-bp of *Hd1*. Position indicates the corresponding location of chromosome 1 in Nipponbare (IRGSP v1.0). Dot indicates the same as Nipponbare reference sequences, "-" indicates deletion in that position. The accession names, subspecies, origin and sequence changes are listed.
(XLSX)

**S9 Table. Selection parameters of five *Hd1* types in several countries.** The *Hd1* gene region and nearby ± 10-kb of traditional landraces were used for calculation.
(DOCX)

**S10 Table. The presence or absence of the 36-bp fragment in 3K landraces, weedy rice and wild rice.**
(XLSX)

**S1 Fig. View of the 36-bp deletion in the *Hd1* gene first exon.** The Integrative Genomics Viewer view of aligned reads of a 162-bp first exon region of *Hd1* gene is shown. The pair read sequencing was performed on an Illumina platform. Alignments toward the Nipponbare IRGSP 1.0 (top row) are represented as gray polygons and mismatched nucleotides as orange, blue, red and green bars. The second row is the alignment of Nipponbare Illumina reads and the third row the alignment of Kasalath reads. Blue arrow at bottom points to the 36-bp deletion region.
(TIF)

**S2 Fig. Geographic distribution of rice subtypes of each Asian country shown in Fig 3.** Colors indicate different subtypes and the pie size indicates the sample size. Only the traditional landraces in the 3K dataset are illustrated. Temperate *japonica*: light blue; tropical *japonica*: green; *indica*: yellow; *aus*: orange; *aromatic*: dark blue; admixture: grey. Because of no information about tradition/modern types for Chinese accessions, all accessions from China are used for illustration. The Asia map was downloaded from https://freevectormaps.com/world-maps/WRLD-EPS-01-0017. Republished from https://freevectormaps.com/world-maps/WRLD- EPS-01-0017 under a CC BY license, with permission from FreeVectorMaps.com, original copyright 2020.
(TIF)

**S3 Fig. The stem-loop structure of the 36-bp deletion located at the first exon in Napponbare.** The mfold package at http://unafold.rna.albany.edu/?q=mfold was used. The free energy (dG) is -2.06 kcal mol-1.
(TIF)

## Acknowledgments

We appreciate the discussion and input from Dr. Chih-Ming Hung on allele network analysis. We thank Ms. Lie-Hong Wu for maintenance of greenhouse plants and Ms. Laura Smales (BioMedEditing, Toronto, Canada) for English editing. The eastern Asia map in Fig 3 and S2 Fig are from https://freevectormaps.com/world-maps/WRLD-EPS-01-0017.

## Author Contributions

**Data curation:** Cheng-Chieh Wu, Fu-Jin Wei, Wan-Yi Chiou, Yuan-Ching Tsai, Hshin-Ping Wu, Dhananjay Gotarkar, Zhi-Han Wei, Ming-Hsin Lai.

**Funding acquisition:** Yue-Ie Caroline Hsing.

**Project administration:** Yue-Ie Caroline Hsing.

**Visualization:** Cheng-Chieh Wu, Wan-Yi Chiou.

**Writing – original draft:** Yue-Ie Caroline Hsing.

**Writing – review & editing:** Yue-Ie Caroline Hsing.

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
