## [Decision Letter · Decision Letter 0]

26 May 2020

PONE-D-20-11893

Studies of rice Hd1 haplotypes worldwide reveal adaptation of flowering time to different environments

PLOS ONE

Dear Dr. Hsing,

Thank you for submitting your manuscript to PLOS ONE. After careful consideration, we feel that it has merit but does not fully meet PLOS ONE’s publication criteria as it currently stands. Therefore, we invite you to submit a revised version of the manuscript that addresses the points raised during the review process. In particular, one of the reviewers criticizes the part of Champa rice because it still needs to be supported by more lines of evidence to reach a solid conclusion. Because the part may not be really necessary for the main story of current manuscript, thus you may reduce or delete it. 

We look forward to receiving your revised manuscript.

Kind regards,

Jong-Seong Jeon, Ph.D.

Academic Editor

PLOS ONE

Journal Requirements:

'The authors have declared that no competing interests exist.'

We note that one or more of the authors are employed by a commercial company: FourOneFour Studio.

3. We note that Figure 2 in your submission contain map images which may be copyrighted.

We require you to either (a) present written permission from the copyright holder to publish this figure specifically under the CC BY 4.0 license, or (b) remove the figures from your submission:

b. If you are unable to obtain permission from the original copyright holder to publish this figure under the CC BY 4.0 license or if the copyright holder’s requirements are incompatible with the CC BY 4.0 license, please either i) remove the figure or ii) supply a replacement figure that complies with the CC BY 4.0 license. Please check copyright information on all replacement figures and update the figure caption with source information. If applicable, please specify in the figure caption text when a figure is similar but not identical to the original image and is therefore for illustrative purposes only.

Reviewers' comments:

Reviewer's Responses to Questions

**Comments to the Author**

1. Is the manuscript technically sound, and do the data support the conclusions?

Reviewer #1: Yes

Reviewer #2: Partly

2. Has the statistical analysis been performed appropriately and rigorously? 

Reviewer #1: Yes

Reviewer #2: No

3. Have the authors made all data underlying the findings in their manuscript fully available?

Reviewer #1: Yes

Reviewer #2: Yes

4. Is the manuscript presented in an intelligible fashion and written in standard English?

Reviewer #1: Yes

Reviewer #2: No

5. Review Comments to the Author

Reviewer #1: The manuscript entitled “Studies of rice Hd1 haplotypes worldwide reveal adaptation of flowering time to different environments” by Wu et al. investigates how the variations in rice Hd1 halotypes contribute to rice domestication and adaptation. This study analyzed the Hd1 gene sequence alterations in various local rice accessions as well as weedy rice and wild rice to find out different hd1 halotypes.

This manuscript is generally well-written and organized in a logical manner. The data presented by these authors support their conclusions. Here are some minor to make it easier for the readers to understand.

Minor points:

1. Page 3, line 67: Early heading date (Ehd1) => Early heading date 1 (Ehd1)

2. Page 3, lines 66-71: Overall, it is confused whether they are gene name or protein name. I understood that the authors meant them gene names(Hd1, Ehd1, Hd3a, and Ghd7). If so the words “which encodes” after each gene names.

3. Pages 6-7, lines 164~176: Simply listing synonyms for the Hd1 gene does not seem to provide that much information. It does not seem bad to make a table and insert references.

4. Page 13, lines 286-287: 31 accessions type 19 hd1 =>31 accessions are type 19 hd1.

5. Page 14, line 315: rufupogon => rufipogon

6. Page 17, line 374 and 375: Hd1 => Hd1

7. Page 22, line 466: japonica => japonica

8. Page 22, line 467: indica => indica

9. Page 23, line 498: includingπ => including π

10. Page 27, line 580: CONSTAINS => CONSTANS, Early heading date => Early heading date 1

11. The table 2 and 3 are a little hard to see and understand. I hope it can be fixed more easily and in a better way.

12. At the beginning of the discussion session, the authors mentioned that photoperiod insensitivity has been beneficial to rice production. However, Endo-Higashi and Izawa (2011) showed that flowering time gene Hd1 could control tiller and panicle development in rice.

Therefore, it might be necessary to mention that the Hd1 gene could be involved in the rice developments, along with controlling of the flowering timing, which affect rice productivity.

Reviewer #2: The manuscript entitled “Studies of rice Hd1 haplotypes worldwide reveal adaptation of flowering time to different environments” describes the relationship between sequence information and flowering time using published 3K rice genome data, accession of weedy rice sequences, and newly determined genome sequences of likely Champa lines in Taiwan. They found three new loss-of-function alleles of Hd1 from the data. Then they found that two of them, type 7 and type 13 alleles, are major ones among loss-of-function Hd1 alleles and have a long history after their selection. More than one-third of the world’s rice accessions contain these two mutations. Interestingly, they distributed among sub-tropical and tropical accsssion.

In addition, they also found that “Champa” lines may not belong to aus, but to the indica cluster. The allele distribution of rice weedy were also examined.

Since Hd1 gene is a major QTL gene to control rice flowering time, it make sense to analyze Hd1 alleles in details. However, there are several related papers already published. Thus, the novelty of this manuscript is not so high. Unfortunaltely, the authors may not recognize a few key papers related to Hd1 haplotypes such as Fujino et al. (2010) and Itoh et al. (2017), thus their discussion should be reorganized with them. I am afraid that the authors just focused on the SNPEffect data of Hd1 among 3K genome too much and missed some important aspect of Hd1 diversity. Especially, haplotype network classification of 3K data are necessary in this manuscript. The analysis of Hd1 introgression as shown in the not-cited papers, instead of selective sweep analysis, such as Tajima’s D would be appropriate in these data.

In addition, it is very funny that 1.9kb insertion in type 19 was detected among 3K data since 3K data use short-reads less than 100 bp and mapped the read into IRGSP1.0. Thus, the authors should describe how they got information on type19 mutations.

Furthermore, they should describe the allele of previously known 36bp deletion in japonica rice in this manuscript. This allele contain the in-frame deletion which caused 12 amino acid from very conserved region of CO like genes though there is no evidence which affect the function of Hd1 before. I am afraid that all this allele information is missing in the public 3K vcf file.

The Champa story and weedy rice story looks extra. They should be independent manuscript if possible.

Finally, when I read this manuscript, I am often confused that which data is new and novel from this work or ones from previous works. The authors should try to clarify this point better.

Minor points,

1) In the introduction section, description of previous analysis on Hd1 haplotypes are largely missing. I am afraid that it may become a biased presentation to pretend the novelty of this work.

2) The previous paper should cited with the first authors. “Shimamoto and coworkers” and “Yoshimura and coworkers” are inappropriate. Only in some reviews, I saw this style before.

3) How they got the flowering time graph in Fig.1 are not well explained yet. If they compared distinct lines containing the same mutation, they should plot all the data including outlier data.

6. PLOS authors have the option to publish the peer review history of their article (what does this mean?). If published, this will include your full peer review and any attached files.

Reviewer #1: Yes: Lae-Hyeon Cho

Reviewer #2: No

---

## [Author Response · Author response to Decision Letter 0]

14 Jul 2020

Reply to reviewers’ comments

Reviewer 1

Thank you very much for the suggestions. I hereby answer accordingly.

Points 1 through 10:

Yes, we changed it according to the comments.

Points 11:

Table 3 was moved to S3 Table in the revision accordingly.

Point 12:

Thank you very much for the suggestion. We added it to the discussion and cited the reference accordingly. Hd1 could have dual effect on rice production, i.e. involving in the rice development and controlling of flowering time.

Reviewer 2:

Thanks a lot for the critical comments and they certainly made improvements on this work. There are many major points so I reply one by one.

1. The part related to Champa rice was indeed a different story and it was removed in this version. 

2. We performed allele network analysis as suggested. Because we only used the land races carrying types 7 and 13, a clear-cut network was shown (Fig. 2 in the current submission). We were able to show the area and types the mutations occurred and how they expanded by human migration. Together we demonstrated that introgressions and human activities played important roles for hd1 diversification/adaptation.

3. Thank you very much for pointing out how we detected 1.9 kb insertion in 3K data. I then double checked with my former PhD student FJW. He indeed wrote a python script 3 years ago and quickly scanned the whole data set followed by IGV confirmation. The method was updated and script provided as Supplementary Material.

4. About the 36-bp insertion in the first exon. We performed the analysis and the results were shown in the revision. Because 36-bp insertion is very short, it is not easy to write a suitable script to scan the whole data set in large/fast scale. Thus, we used visualization identification as mentioned in the revision. Of the 1070 traditional landraces in 3K, all indica and tropical japonica accessions had this short insertion. Only few temperate japonica accessions were the same as Nipponbare, including 10 accessions from Korea, 6 from Japan, 4 from China, and 1 from Taiwan. For the 6 Japanese ones, 4 are real landraces (from NARO Genebank website), 1 is Nipponbare, and the other CX330 Yueguang. It took us a while to figure out this Yueguang should be Koshihikari as its Han characters (Kanji) pronounced as Yueguang in Chinese. It was picked up by CAAS and maybe even IRRI breeders do not know its real name is Koshihikari.

All 15 Asian wild rice tested also have this 36-bp fragment, however, 3 of them have identical or similar seq SNP with Nipponbare. We suggest the resource could be Japanese accession with 36-bp fragment. We do not have enough sequence information to check all though.

5. About which data was new and novel or which ones from previous work. Yes, we improve it accordingly. Thank you very much for pointing out.

Minor points

1. We added more discussion accordingly.

2. Yes, it was improved in the current session.

3. For the ~2500 accessions with seeds in IRRI germplasm, IRRI scientists performed a series of phenotyping analyses and data were available in SNP-SEEK, as mentioned in the ms. We downloaded these data and ran analysis. “Wild type” means the accessions without any LOF mutation and thus was used as outliers. Wild type accessions had the average heading date of 106 days, and the other types ranged from 77 to 93 days (S4 Table). The accession numbers used for WT, type 3, 7, 12, 13, 19, 20, 21 are 1263, 19, 213, 21, 552, 16, 16, and 3, respectively. We hope these data were enough to confirm they were LOF mutations.

4. There are 10 supplementary tables in the current version. For instances, we provided the SNPs information of hd1 gene region of landraces carrying type 7 and 13 mutations, we also listed the accessions belong to each haplotype, etc. Because there are around 400 landraces consisting types 7 and 13 mutations, they were shown as Excel files in Supplementary tables.

---

## [Decision Letter · Decision Letter 1]

14 Aug 2020

PONE-D-20-11893R1

Studies of rice Hd1 haplotypes worldwide reveal adaptation of flowering time to different environments

PLOS ONE

Dear Dr. Hsing,

Thank you for submitting your manuscript to PLOS ONE. Both reviewers and I feel that this revised manuscript deserves PLOS ONE’s publication. Before its publciation we invite you to revise only few minor points raised during the review process.

We look forward to receiving your revised manuscript.

Kind regards,

Jong-Seong Jeon, Ph.D.

Academic Editor

PLOS ONE

Reviewers' comments:

Reviewer's Responses to Questions

**Comments to the Author**

1. If the authors have adequately addressed your comments raised in a previous round of review and you feel that this manuscript is now acceptable for publication, you may indicate that here to bypass the “Comments to the Author” section, enter your conflict of interest statement in the “Confidential to Editor” section, and submit your "Accept" recommendation.

Reviewer #1: All comments have been addressed

Reviewer #2: (No Response)

2. Is the manuscript technically sound, and do the data support the conclusions?

Reviewer #1: Yes

Reviewer #2: Yes

3. Has the statistical analysis been performed appropriately and rigorously? 

Reviewer #1: Yes

Reviewer #2: Yes

4. Have the authors made all data underlying the findings in their manuscript fully available?

Reviewer #1: Yes

Reviewer #2: Yes

5. Is the manuscript presented in an intelligible fashion and written in standard English?

Reviewer #1: No

Reviewer #2: Yes

6. Review Comments to the Author

Reviewer #1: Thank you for revising the manuscript. Most of my comments have been well revised. However, I still have some minor issues.

Arabidopsis CO => CO(many times in the manuscript)

Page 4, line 92; page 5, line 101 : orthologue => ortholog

Page 14, line 295 : 446 bp and 452 bp => 446 and 452 bp

Page 16, line 327 : Guangdona => Guangdong(?)

Page 19, Line 422 : Hd1 gene => Hd1

Reviewer #2: This version of manuscript has been much improved since the authors have addressed my previous comments politely.

Minor comments;

In Table3, each circle may contain all subgroups in 3K. Thus, similar data for only a single subgroup such as aus or temperate japonica should be presented as a supplemental figure to imply that local requirement of the Hd1 function may not be affected by the subgroup difference.

Explanation of Tajima'D results may not be restricted to selective sweeps. Thus, the authors should also describe other possibility such as population size changes (or bottle neck effect).

7. PLOS authors have the option to publish the peer review history of their article (what does this mean?). If published, this will include your full peer review and any attached files.

Reviewer #1: **Yes: **Lae-Hyeon Cho

Reviewer #2: No

---

## [Author Response · Author response to Decision Letter 1]

27 Aug 2020

We hereby submit the 2nd revision for our previous manuscript entitled “Studies of rice Hd1 haplotypes worldwide reveal adaptation of flowering time to different environments”. Thanks a lot to the reviewers, and we modified the original manuscript accordingly.

Here are the points-to-points replies for the comments:

Reply to Reviewer #1:

1. Arabidopsis CO => CO

Yes, we changed them accordingly. Thank you very much.

2. typos or mistakes in pages 4, 14, 16 and 19

Yes, we changed them accordingly. Thank you very much.

Reply to Reviewer #2:

1. About the display of these subgroups (Fig 3). Yes, we added S2 Fig in the supplementary materials. It indeed improves the interpretation of Fig 3. Thank you very much. 

2. About explanation of Tajima’s D results. Thank you very much for the reminding. We added a short paragraph for the interpretation accordingly.

---

## [Editor Report · Decision Letter 2]

31 Aug 2020

Studies of rice Hd1 haplotypes worldwide reveal adaptation of flowering time to different environments

PONE-D-20-11893R2

Dear Dr. Hsing,

We’re pleased to inform you that your manuscript has been judged scientifically suitable for publication and will be formally accepted for publication once it meets all outstanding technical requirements.

Kind regards,

Jong-Seong Jeon, Ph.D.

Academic Editor

PLOS ONE
---

## [Editor Report · Acceptance letter]

7 Sep 2020

PONE-D-20-11893R2 

Studies of rice *Hd1* haplotypes worldwide reveal adaptation of flowering time to different environments 

Dear Dr. Hsing:

I'm pleased to inform you that your manuscript has been deemed suitable for publication in PLOS ONE. Congratulations! Your manuscript is now with our production department. 

Kind regards, 

on behalf of

Professor Jong-Seong Jeon 

Academic Editor

PLOS ONE